# Mid-trimester cervical length not associated with HIV status among pregnant women in Botswana

Ingrid Liff[1¤a]*, Rebecca Zash[2,3], Denis Mingochi[4], Findo Tsaone Gaonakala[3], Modiegi Diseko[3], Gloria Mayondi[3], Katherine Johnson[5], Kaitlyn James[1], Joseph Makhema[3], Roger Shapiro[2,3], Blair J. Wylie[1¤b]

1 Division of Maternal-Fetal Medicine, Department of Obstetrics and Gynecology, Massachusetts General Hospital, Boston, MA, United States of America, 2 Department of Infectious Disease, Beth Israel Deaconess Medical Center, Boston, MA, United States of America, 3 Botswana Harvard AIDS Institute Partnership, Gaborone, Botswana, 4 Scottish Livingstone Hospital, Molepolole, Botswana, 5 Division of Maternal-Fetal Medicine, Department of Obstetrics and Gynecology, Beth Israel Deaconess Medical Center, Boston, MA, United States of America

¤a Current address: Division of Maternal Fetal Medicine at Tufts Medical Center, Boston, MA, United States of America
¤b Current address: Division of Maternal Fetal Medicine at Beth Israel Deaconess Medical Center, Boston, MA, United States of America
* iliff@tuftsmedicalcenter.org

## Abstract

### Objective

HIV-infected women on antiretroviral therapy have a higher risk of preterm birth than HIV-uninfected women in Botswana. To better understand the mechanism for preterm birth among HIV-infected women, we evaluated whether mid-trimester cervical length differed by HIV status as cervical shortening is associated with an increased risk for preterm birth.

### Methods

We conducted a prospective cohort study among pregnant women receiving care at the Scottish Livingstone Hospital in Molepolole, Botswana. Consecutive women referred for routine obstetrical ultrasound were consented and enrolled if between 22w0d and 24w6d by ultrasound biometry. Blinded to maternal HIV status, an obstetrician measured transvaginal cervical length using standardized criteria. Cervical length, as well as the proportion of women with a short cervix (<25mm), were compared among HIV-infected and HIV-uninfected women. The acceptability of transvaginal ultrasound was also evaluated.

### Results

Between April 2016 and April 2017, 853 women presenting for obstetric ultrasound were screened, 187 (22%) met eligibility criteria, and 179 (96%) were enrolled. Of those enrolled, 50 (28%) were HIV-infected (86% on antiretroviral therapy), 127 (71%) were HIV-uninfected, and 2 (1%) had unknown HIV status. There was no significant difference in mean cervical length between HIV-infected and HIV-uninfected women (32mm vs 31mm, p =

**Data Availability Statement:** All relevant data are within the manuscript and its Supporting Information files.

**Funding:** This work was supported by a grant from the Queenan Fellowships for Global Health sponsored by the Foundation for SMFM (IL). https://foundationforsmfm.org. The funders had no role in study design, data collection and analysis, decision to publish, or preparation of the manuscript.

**Competing interests:** The authors have declared that no competing interests exist.

0.21), or in the proportion with a short cervix (10% vs 14%, p = 0.44). Acceptability data was available for 115 women who underwent a transvaginal ultrasound exam. Of these, 112 of 115 (97%) women deemed the transvaginal scan acceptable.

## Conclusions

The increased risk of preterm birth observed among HIV-infected women receiving antiretroviral therapy in Botswana is unlikely associated with mid-trimester cervical shortening. Further research is needed to understand the underlying mechanism for preterm birth among HIV-infected women.

## Introduction

Antiretroviral therapy (ART) during pregnancy is necessary for both maternal health and reduction of mother-to-child transmission of HIV (MTCT)[1–4], but evidence shows increased risk of preterm birth (PTB) in HIV-infected women on ART during pregnancy– both in the high- and low- income settings.[5–19] This risk has been observed for multiple ART regimens and is highest among women on ART from conception.[6,16,20] The etiology of PTB among women on ART is unexplained, leading to a lack of potential interventions to reduce preterm birth in this population.[21]

Cervical length is a strong predictor of risk for PTB. Research from the United States of America (USA) shows that a cervical length below the 5$^{th}$ percentile (22mm) at 24 weeks of gestation increases the risk of preterm birth 10-fold,[22] and treating these at-risk women with vaginal progesterone decreases the risk of preterm delivery by almost 50%.[23–26] The specific mechanism of action of progesterone in prevention of preterm birth is unknown,[27] although it is hypothesized that vaginal progesterone may have an immunomodulatory effect that slows cervical shortening.[28]

PTB among HIV-infected women on ART may be mediated through cervical shortening due to abnormalities in the physiologic immune environment. During pregnancy, there is a physiologic shift from Th1- to Th2-mediated cytokine activity[29] that is necessary to maintain pregnancy;[30] ART reverses this shift.[31] Progesterone, on the other hand, supports the immunologic shift from Th1- to Th2,[32,33] and treatment with vaginal progesterone supports Th2 activity at the level of the cervix.[28] It is therefore plausible that progesterone could function as an immune modulator that prevents PTB in HIV-infected women, especially those on ART.

In Botswana, ~25% of pregnant women are HIV-infected, with high levels of antenatal care (>95%) and ART uptake (>90%). Recent data from Botswana shows that PTB is more common among HIV-infected women than HIV-uninfected women (22.5% vs. 15.6%, aRR 1.39, 95% CI 1.33–1.96).[17] The Botswana Harvard AIDS Institute Partnership (BHP) has been conducting HIV research in pregnancy for more than 20 years. Leveraging the existing infrastructure of a large birth surveillance study run through BHP (NIH/NICHD R01 HD080471, Shapiro PI), we explored the relationship between mid-trimester cervical length and HIV infection.

## Materials and methods

This cervical length study was nested within a large birth surveillance study (NIH/NICHD R01 HD080471, Shapiro PI) collecting data at 8 large delivery sites in Botswana, described

previously.[17] This larger study collects data from the maternal health records at the time of delivery for all women (regardless of HIV status), covering approximately 45% of all births in the country. Between April 2016 and April 2017, we prospectively enrolled pregnant women presenting for ultrasound examination at one of these sites, Scottish Livingstone Hospital (SLH) in Molepolole. Cervical length study participants were assigned a unique study number allowing linkage to the larger birth surveillance study collecting data at delivery.

Consecutive women referred for routine obstetrical ultrasounds (recommended by prenatal care guidelines in Botswana) were approached for participation in the study by a research assistant. Written informed consent was obtained in Setswana or English, as appropriate. Women included in the study were at least 18 years old, were carrying a live singleton gestation between 22w0d and 24w6d, and planned to deliver at SLH. The gestational age window for inclusion mirrored the initial paper on the association between cervical length and preterm birth.[22] Women with a multiple gestation, suspected fetal anomalies, fetal demise, or active symptoms of preterm labor (e.g. contractions, vaginal bleeding, or loss of fluid) were excluded from participation.

### Ultrasound examination and cervical length measurements

A maternal-fetal medicine physician from the USA (IL) performed all abdominal and transvaginal ultrasound exams, and was blinded to maternal HIV status. Gestational age was assigned from biometric measurements of the biparietal diameter, head circumference, abdominal circumference, and femur length. Each biometric parameter was measured thrice and the average used to assign gestational age using the Hadlock formula.[34]

The cervical length was measured by transvaginal ultrasound using standardized criteria established by the Perinatal Quality Foundation.[35] Using the best of three consistent measurements, the cervical canal was measured from the internal os to the external os in the mid-sagittal plane. For each image, the cervix occupied at least 75% of the screen, the bladder area was visible, and the anterior and posterior portions of the cervix were equal. Each cervix was measured over at least three minutes to evaluate for dynamic change, and fundal pressure was applied at the end of each exam to evaluate for possible change in cervical length with pressure.

The majority of ultrasound exams were performed using a refurbished portable Sonosite™ Titan model fitted with a transvaginal probe. This ultrasound machine was procured through the Soundcare program. Other ultrasound machines also were used while integrating study procedures at SLH, including Mindray Diagnostic Ultrasound System Model DC-N3 and Sonoscape Model SS1-4000, owned by Scottish Livingstone Hospital. Compatible transvaginal probes were available for each machine. Adequate imaging was obtained with each machine.

The transvaginal probe was cleaned prior to each exam with a 0.55% ortho-phthaladehyde solution. Sterile probe covers were used with each separate exam, along with sterile gel externally.

An ultrasound report was prepared for each patient after the exam and included in the patient's medical record. If the cervix measured less than 15mm, the patient was counseled about the signs and symptoms of preterm labor. Women with a history of preterm birth and identified cervical shortening were referred to the Princess Marina Hospital in Gaborone.

After the cervical length measurement, each patient completed a questionnaire with information collected about the last menstrual period, socioeconomic status, obstetric history, HIV history including ART regimen if HIV-infected, and questions regarding acceptability of transvaginal ultrasound. Reported HIV-infected status was confirmed by chart review by the research assistant at the time of the questionnaire.

Through linkage with the larger birth outcomes study, we also collected additional maternal demographics, maternal medical and obstetric history, medications received during pregnancy, date of delivery, details of infant status at delivery, and infant outcome through 28 days of life. Additionally, for HIV-infected women we collected date of HIV diagnosis, antiretroviral treatment regimens and CD4 cell count in pregnancy. Gestational age at delivery was determined by study biometry.

## Statistical analysis

We calculated the power analysis for this study based on an expected difference of 5mm between mean cervical length in HIV-infected and HIV-uninfected women. For HIV-uninfected women, we predicted a mean cervical length of 35mm and a standard deviation of 8mm based on the established USA cohort.[22] We hypothesized the standard deviation would be wider (12mm) among HIV-infected women. For 50 HIV-infected women and at least 100 HIV-uninfected women, we had >80% power to detect a difference in mean cervical length of 5mm.

Mean cervical length was compared between HIV-infected and HIV-uninfected women using the student's t-test. We performed an exploratory analysis using Pearson's chi-squared test to compare the proportion of HIV-infected women with short cervix (cervical length <25mm) to the proportion of HIV-uninfected women with short cervix. In addition, multivariable logistic regression models were employed to explore the relationship between short cervix (<25mm), HIV infection status, history of preterm birth, and socioeconomic status. We also performed an exploratory analysis to compare mean mid-trimester cervical length in Botswana women with mean mid-trimester cervical length in USA women using Kolmogorov-Smirnov equality-of-distributions test. P-values <0.05 were considered statistically significant. All statistical analyses were conducted in Stata 14.0 (College Station, TX: StataCorp).

## Ethics approvals

Prior to voluntary participation in the study, each woman signed a written consent form in Setswana or English, per her preference. The study protocol and consent forms were approved by the Human Research Development Council (HRDC) in Botswana, the Institutional Review Board at Massachusetts General Hospital, and the Ethics Committee at the Scottish Livingstone Hospital prior to the start of the study.

## Results

Between April 2016 and April 2017, 853 women were screened with an ultrasound exam for viability, fetal number, gross anomalies, amniotic fluid, and biometry. Of those screened, 190 (22%) women were at 22-24w6d GA, and 187 (98%) of these women met eligibility criteria. These women were approached for participation and 179 women (96%) were enrolled into the study (8 declined participation). Our study population included 127 (71%) HIV-uninfected women and 50 (28%) HIV-infected women. There were 2 patients with unknown HIV status.

Forty-two (84%) of all HIV-infected women were on ART at the time of recruitment, and 30 (60%) of all HIV-infected women were on ART at the time of conception. ART regimens included efavirenz / emtricitabine / tenofovir (EFV-based, 51%), nevirapine / lamivudine / zidovudine (NVP-based, 27%), dolutegravir / emtricitabine / tenofovir (DTG-based, 12%), or other (10%). CD4 cell counts and HIV RNA were not routinely available.

Table 1 describes the characteristics of our population. HIV-infected women were older, less likely to be nulliparous, and had lower educational attainment compared with HIV-

**Table 1. Maternal demographics.**

| | Total cohort (n = 179) | HIV-uninfected (n = 127) | HIV-infected (n = 50) | p-value |
|---|---|---|---|---|
| *Demographics* | | | | |
| Mean age (mean ± SD) | 28 ± 6.8 | 26 ± 6.2 | 32.5 ± 6 | <0.01 |
| Mean gestational age at CL measurement | 22.8 ± 0.9 | 22.8 ± 0.9 | 22.8 ± 0.9 | 0.92 |
| Education > primary | 147 (83%) | 112 (88%) | 34 (68%) | <0.01 |
| *Employment* | | | | 0.17 |
| Salaried job | 50 (28%) | 34 (26%) | 15 (28%) | |
| Student | 5 (3%) | 5 (4%) | 0 (0%) | |
| Housework | 119 (66%) | 84 (66%) | 34 (68%) | |
| Other | 5 (3%) | 4 (1.5%) | 1 (4%) | |
| *Language spoken at home* | | | | 0.62 |
| Setswana | 174 (97.2%) | 124 (98%) | 48 (96%) | |
| Other | 5 (3%) | 3 (2%) | 2 (4%) | |
| *Co-morbidities* | | | | |
| History of alcohol use | 1 (<1%) | 0 | 1 (2%) | 0.11 |
| History of malaria | 1 (<1%) | 0 | 1 (2%) | 0.11 |
| History of tuberculosis (ever) | 10 (5%) | 5 (4%) | 5 (10%) | 0.12 |
| *TB currently* | 1 (<1%) | 1 (<1%) | 0 | 0.53 |
| Other chronic disease | 6 (3%) | 5 (4%) | 1 (2%) | 0.47 |
| *Obstetric history* | | | | |
| *Parity* | | | | <0.01 |
| Nulliparous | 60 (33.5%) | 53 (42%) | 6 (12%) | |
| Parity >1 | 119 (66.5%) | 74 (58%) | 44 (88%) | |
| History of preterm delivery | 3 (2%) | 2 (2%) | 1 (2%) | 0.86 |
| History of miscarriage or termination | 15 (8%) | 7 (6%) | 5 (14%) | 0.18 |
| History of low birth weight infant (<2500gm) | 15 (8%) | 6 (5%) | 9 (18%) | <0.01 |
| History of cervical surgery | 1 (<1%) | 1 (<1%) | 0 (0%) | 0.57 |
| *Prenatal care* | | | | 0.55 |
| Initiated prenatal care < = 20w GA | 171 (96%) | 122 (96%) | 47 (94%) | |
| Initiated prenatal care >20w GA | 8 (4%) | 5 (4%) | 3 (6%) | |

uninfected women, but there were no differences between the HIV-infected and uninfected women with regards to history of preterm birth, substance use or prenatal care uptake.

## Cervical length

The mean cervical length in the overall population was 31mm (range 9.8-46mm). There was no difference in mean cervical length between HIV-infected and HIV-uninfected women (32mm vs. 31mm, p = 0.21) (Table 2). The distribution of cervical length was similar between the groups (Fig 1).

**Table 2. Cervical length by HIV status.**

| Cervical Length (mm) | Total Cohort (n = 179) | HIV-uninfected (n = 127) | HIV-infected (n = 50) | p-value |
|---|---|---|---|---|
| Mean cervical length (mean ± SD) | 31.4 ± 6.3 | 31.1 ± 6.4 | 32.4 ± 6.1 | 0.22 |
| Median cervical length (median (IQR) | 31.8 (28.5–35) | 31.7 (27.2–34.8) | 32.2 (29–36.3) | 0.29 |
| Cervical length < 15mm (N[%]) | 4 (2%) | 3 (2%) | 1 (2%) | 0.88 |
| Cervical length < 25mm (N[%]) | 24 (13%) | 19 (15%) | 5 (10%) | 0.39 |

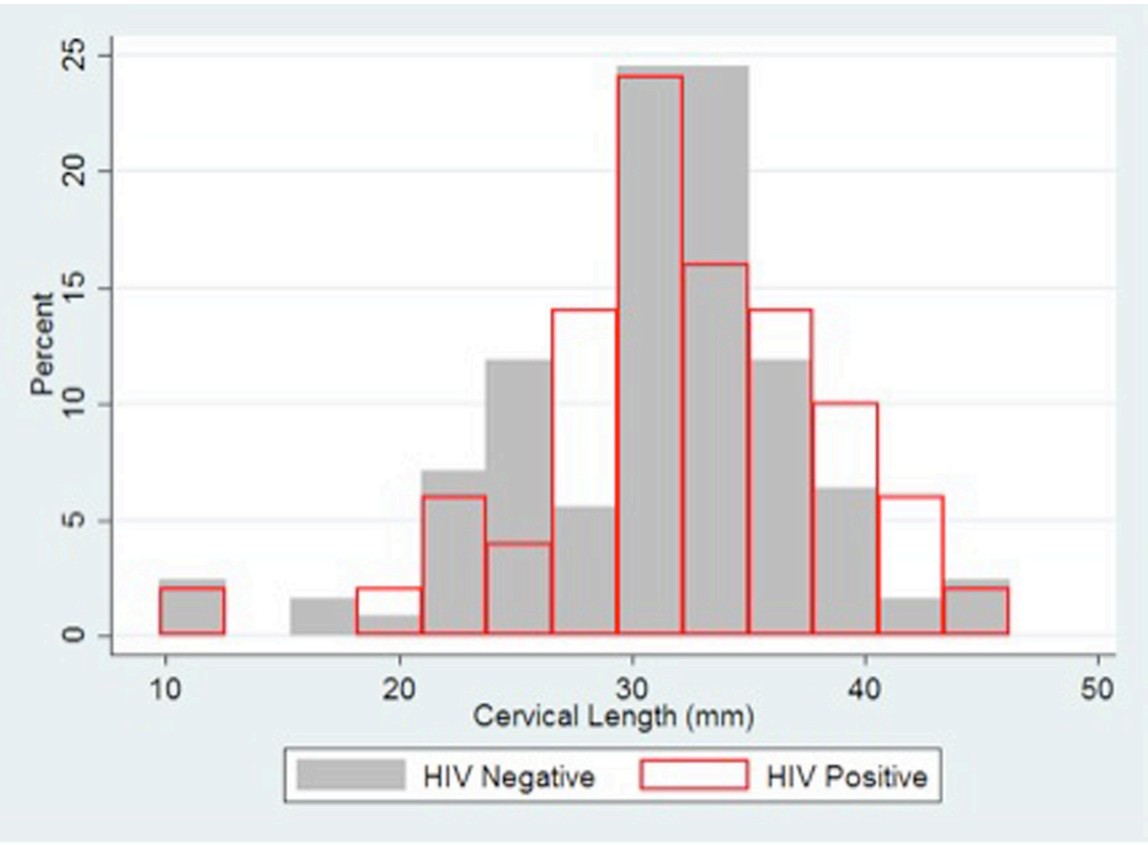

**Fig 1. Distribution of cervical length by HIV status.**

The proportion of women with short cervix (<25mm) (10% HIV-infected vs 14% uninfected, p = 0.44) did not differ by HIV-status, including after controlling for history of preterm delivery and socioeconomic status (aOR, 0.61; 95% CI, 0.21–1.77; p = 0.34). There were only 4 (2%) women in our study who had CL less than 15mm (one was HIV-infected). Twenty-four women (13%) had a cervical length less than 25mm; these women were younger than women with cervical length greater than or equal to 25mm (24.3 yrs vs 28.6 yrs, p = <0.01).

Among HIV-infected women, women on ART at conception (n = 30, 60%) had a similar mean cervical length compared with women who started ART during pregnancy (n = 7, 14%) or were not on ART at the time of recruitment (n = 7, 14%) (32mm vs 35mm, p = 0.13) (Table 2A). Five women on ART had unknown timing of ART, and one woman was unknown whether she was on ART. Cervical length was similar among women on different ART regimens (31mm on EFV-based ART (N = 21), 33mm on DTG-based ART (N = 5), 35mm on NVP-based ART (N = 11), and 31mm on LPV-r based ART (N = 4). One patient on ART had an unknown regimen.

**Table 2A. Cervical length by HIV status for those HIV-infected women on ART at the time of conception.**

| Cervical Length (mm) | HIV-uninfected (n = 127) | HIV-infected on ART at conception (n = 30) | p-value |
|---|---|---|---|
| Mean cervical length (mean ± SD) | 31.1 ± 6.4 | 32.1 ± 5.2 | 0.41 |
| Median cervical length (median (IQR) | 31.7 (27.2–34.8) | 32 (29–35) | 0.62 |
| Cervical length < 15mm (N[%]) | 3 (2%) | 0 (0%) | 0.39 |
| Cervical length < 25mm (N[%]) | 19 (15%) | 3 (10%) | 0.48 |

## Pregnancy outcomes

Pregnancy outcomes were captured for 122 of 179 women (68% overall; 30 HIV-infected women, 25%; 92 HIV-uninfected women, 75%). Study biometry was used to calculate gestational age on the date of delivery. In this cohort, a total of eight women delivered preterm (6.6% overall). There was no difference in the preterm birth rate among those women with HIV infection (2/30 HIV-infected 6.7%) compared to those without HIV infection (6/92, 6.5%; p = 0.59). Of the preterm deliveries, six of the 8 women delivered in the 36[th] week, and two of the 8 women delivered prior to 36 weeks GA (range 26–36 weeks GA). One HIV-infected woman delivered in the 26[th] week (mid-trimester CL 36mm), and one HIV-uninfected woman delivered in the 34[th] week (mid-trimester CL 32mm). Among all women with preterm delivery in this cohort, only 1 patient (HIV-uninfected) had a mid-trimester cervical length less than 25mm (22mm). Of the women in the study with available outcomes data (n = 122), thirteen women had CL <25mm (range 11-24mm). One woman with CL<25mm delivered in the 36[th] week, and the remaining 12 of 13 women with short cervix delivered at >37 weeks (range 37–42 weeks GA).

## Acceptability of transvaginal imaging

Acceptability data was available for 115 women who underwent transvaginal ultrasound. For almost all of the women (114/115, 99%), this was their first transvaginal scan. Most women deemed the transvaginal scan acceptable (112/115, 97%). Six (5%) of the 115 women thought TVUS was painful (0/6 women were HIV-infected). The majority of women (112/115 women, 97%) said they would undergo transvaginal scanning again if needed, and 113 of 115 (98%) women said they would recommend transvaginal scanning to a friend if indicated.

## Discussion

To better understand the mechanism for the observed increased risk of PTB among HIV-infected women on ART, we performed a study of mid-trimester cervical length by HIV status. We did not identify a difference in cervical length between HIV-infected women on ART and HIV-uninfected women, nor did we find an increase in the risk for short cervix (<25mm) among HIV-infected women. Our results suggest the risk for PTB among HIV-infected women is not mediated by cervical shortening.

Compared with an established USA cohort, mean cervical length in Botswana was shorter (31mm vs. 35mm, p = <0.001).[22] Compared to cohorts from Africa, mean cervical length in Botswana may be shorter as well. Mean cervical length in a South African cohort in the mid trimester was 33mm.[36] In a recent Zambian cohort, median cervical length was 36mm.[37] Further research is needed to understand whether mid-trimester cervical length differs in distinct settings around the world and whether this has implications for the incidence of PTB.

In Africa, data on acceptability of transvaginal ultrasound are limited, and the results are varied.[38,39] Our results are reassuring that transvaginal ultrasonography is acceptable to women in Botswana, and that with informed consent, transvaginal ultrasound can be used for future studies and in clinical practice in Botswana.

There were many strengths of our study. We did not exclude women based on parity or history of preterm birth, so our results are generalizable for all women in Botswana. At the time of cervical length measurement, the sonographer was blinded to maternal HIV status, limiting bias in cervical length assessment. Gestational age assessment was performed by ultrasound exam at the time of recruitment, limiting variation in gestational age dating methods. All cervical lengths were measured by transvaginal ultrasound, the gold standard for cervical length measurement. The cervical length measurements were performed in a narrow gestational age

window, limiting variation by gestational age and broadening the generalizability of our results.

Our study had several limitations. Due to a limited number of HIV-infected women included in our study, and unavailable CD4 cell count and HIV RNA data for some participants, we were unable to explore differences in cervical length based on ART regimen, timing of initiation of ART in pregnancy, CD4 cell count, or HIV RNA. We also were not able to explore differences in cervical length by history of preterm birth or low socioeconomic status among HIV-infected women.

Our group has previously reported on the increased risk of preterm birth among pregnant women infected with HIV in our population in Botswana.[8,9,16,17] The current study was designed to evaluate the cervical length distributions between HIV-infected and HIV-uninfected women rather than demonstrate a link between cervical shortening and preterm birth as has been previously established.[22] We did not observe a difference in the cervical length distributions by HIV status. While cervical shortening heightens the risk for preterm birth, not all preterm births are preceded by cervical shortening, as was the case for our participants. Among the eight women with preterm births in our cohort, only one had a mid-trimester cervical length less than 25mm. Through additional post hoc power analysis, we determined that an increase in our sample size by 50% in each group would yield similar results.

In conclusion, our study suggests that the increased risk of PTB observed among HIV-infected women in Botswana is unlikely to be associated with mid-trimester cervical shortening. Further research is needed to understand the mechanism of PTB in HIV-infected women receiving ART.

## Supporting information

**S1 Data.**
(XLSX)

## Author Contributions

**Conceptualization:** Ingrid Liff, Rebecca Zash, Roger Shapiro, Blair J. Wylie.

**Data curation:** Ingrid Liff, Denis Mingochi, Findo Tsaone Gaonakala, Katherine Johnson.

**Formal analysis:** Ingrid Liff, Kaitlyn James.

**Funding acquisition:** Ingrid Liff.

**Investigation:** Ingrid Liff, Rebecca Zash.

**Methodology:** Ingrid Liff, Rebecca Zash, Roger Shapiro, Blair J. Wylie.

**Project administration:** Findo Tsaone Gaonakala, Modiegi Diseko, Gloria Mayondi.

**Supervision:** Joseph Makhema, Roger Shapiro, Blair J. Wylie.

**Writing – original draft:** Ingrid Liff.

**Writing – review & editing:** Rebecca Zash, Roger Shapiro, Blair J. Wylie.

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
