## [Decision Letter · Decision Letter 0]

26 Nov 2019

PONE-D-19-28835

Mid-trimester cervical length not associated with HIV status among pregnant women in Botswana

PLOS ONE

Dear Dr. Liff,

Thank you for submitting your manuscript to PLOS ONE. After careful consideration, we feel that it has merit but does not fully meet PLOS ONE’s publication criteria as it currently stands. Therefore, we invite you to submit a revised version of the manuscript that addresses the points raised during the review process.

Please consider the comments of reviewer 1 and the following:

The authors assess cervical length in HIV infected women compared to controls. 

1. Are the outcomes of the pregnancy (maternal) available? If yes, provide information about the gestational age of delivery and the numbers delivering <37 weeks. 

2. Table 2 and 2a: The length <30 mm is not needed since >25 mm is considered normal. 

3. The information about ART can be in the text of the manuscript and not needed in Table 1.

We would appreciate receiving your revised manuscript by Jan 10 2020 11:59PM. To enhance the reproducibility of your results, we recommend that if applicable you deposit your laboratory protocols in protocols.io, where a protocol can be assigned its own identifier (DOI) such that it can be cited independently in the future. For instructions see: http://journals.plos.org/plosone/s/submission-guidelines#loc-laboratory-protocols

We look forward to receiving your revised manuscript.

Kind regards,

David J. Garry, DO

Academic Editor

PLOS ONE

Journal Requirements:

1. Please amend your current ethics statement to address the following concerns:  

a) Did participants provide their written or verbal informed consent to participate in this study?

Additional Editor Comments (if provided):

The authors assess cervical length in HIV infected women compared to controls.

1. Are the outcomes of the pregnancy (maternal) available? If yes, provide information about the gestational age of delivery and the numbers delivering <37 weeks.

2. Table 2 and 2a: The length <30 mm is not needed since >25 mm is considered normal.

3. The information about ART can be in the text of the manuscript and not needed in Table 1.

Reviewers' comments:

Reviewer's Responses to Questions

**Comments to the Author**

1. Is the manuscript technically sound, and do the data support the conclusions?

Reviewer #1: Partly

2. Has the statistical analysis been performed appropriately and rigorously? 

Reviewer #1: Yes

3. Have the authors made all data underlying the findings in their manuscript fully available?

Reviewer #1: Yes

4. Is the manuscript presented in an intelligible fashion and written in standard English?

Reviewer #1: Yes

5. Review Comments to the Author

Reviewer #1: This manuscript represents an excellent study attempting to determine the rate of midtrimester cervical shortening among women with HIV in Botswana. The authors propose to use this evaluation as a proxy measure for understanding the underlying mechanism for an increased preterm birth rate among women with HIV on ART. The scientific design was reasonably rigorous for the question asked, and the manuscript is well prepared.

My chief issue with this manuscript is that there is no data reported regarding whether any of the subjects in the trial actually went on to have a preterm birth. This data may be easily obtainable since this is a nested study within a large longitudinal cohort. Alternatively, because women with short cervix were referred elsewhere, this data is unavailable. It is possible that the authors do not present this information because the purpose of the study was to determine the rate of cervical shortening, rather than the correlation between cervical shortening and preterm birth in this population. However, given that they are speculating at a cause of preterm birth among women with HIV, their relatively small sample population may simply have missed all the women who actually went on to preterm birth. In this case, they are unable to conclude that cervical shortening does not contribute to preterm birth.

The manuscript would be strengthened by reporting gestational age at birth for the cohort. If this population did indeed have an earlier GA at birth among women with HIV, then the authors conclusions are substantiated. If this data is unavailable, the discussion should include a more robust description of the limitations of this population sample in determining contributors to preterm birth. The sentence currently included in the discussion, "Our study was not powered to show a difference in PTB by exposure categories in the cohort, including differences between HIV-infected and HIV-uninfected women." is insufficient in demonstrating to the reader these nuances.

Additional considerations:

- Commentary regarding mechanism of progesterone use in HIV infected women with normal cervical length appears out of place in this manuscript as this is an ultrasound study, not a therapeutic study. If the subjects in the trial did get vaginal progesterone and the authors think this confounded the preterm birth rate in the cohort, that should be explicitly said. Otherwise I would remove this section.

- The acceptability data is interesting, and should be highlighted in the abstract if there is word count to support it.

6. PLOS authors have the option to publish the peer review history of their article (what does this mean?). If published, this will include your full peer review and any attached files.

Reviewer #1: No

---

## [Author Response · Author response to Decision Letter 0]

6 Jan 2020

1. Are the outcomes of the pregnancy (maternal) available? If yes, provide information about the gestational age of delivery and the numbers delivering <37 weeks. 

Reviewer #1 full comment: “This manuscript represents an excellent study attempting to determine the rate of midtrimester cervical shortening among women with HIV in Botswana. The authors propose to use this evaluation as a proxy measure for understanding the underlying mechanism for an increased preterm birth rate among women with HIV on ART. The scientific design was reasonably rigorous for the question asked, and the manuscript is well prepared.

My chief issue with this manuscript is that there is no data reported regarding whether any of the subjects in the trial actually went on to have a preterm birth. This data may be easily obtainable since this is a nested study within a large longitudinal cohort. Alternatively, because women with short cervix were referred elsewhere, this data is unavailable. It is possible that the authors do not present this information because the purpose of the study was to determine the rate of cervical shortening, rather than the correlation between cervical shortening and preterm birth in this population. However, given that they are speculating at a cause of preterm birth among women with HIV, their relatively small sample population may simply have missed all the women who actually went on to preterm birth. In this case, they are unable to conclude that cervical shortening does not contribute to preterm birth.

The manuscript would be strengthened by reporting gestational age at birth for the cohort. If this population did indeed have an earlier GA at birth among women with HIV, then the authors conclusions are substantiated. If this data is unavailable, the discussion should include a more robust description of the limitations of this population sample in determining contributors to preterm birth. The sentence currently included in the discussion, "Our study was not powered to show a difference in PTB by exposure categories in the cohort, including differences between HIV-infected and HIV-uninfected women." is insufficient in demonstrating to the reader these nuances.”

Response 1: Outcomes data was available for only 68% of our cohort, and we included the details in the revised manuscript as suggested. In this subset of women, eight out of 122 women had a preterm birth (6.6% overall; 2/30 HIV-infected women, 6.7%; 6/92 HIV-uninfected women, 6.5%; p=0.59), and one of these 8 women had a short cervix <25mm. These are interesting results, but our ability to draw significant conclusions from this data is limited.

Our group has previously reported on the increased risk of preterm birth among pregnant women infected with HIV in our population in Botswana.1–4 The current study was designed to evaluate the cervical length distributions between HIV-infected and HIV-uninfected women rather than demonstrate a link between cervical shortening and preterm birth as has been previously established.5 We did not observe a difference in the cervical length distributions by HIV status. While cervical shortening heightens the risk for preterm birth, not all preterm births are preceded by cervical shortening, as was the case for our participants. Among the eight women with preterm births in our cohort, only one had a mid-trimester cervical length less than 25mm. Through additional post hoc power analysis, we determined that an increase in our sample size by 50% in each group would yield similar results. In conclusion, our study suggests that the increased risk of PTB observed among HIV-infected women in Botswana is unlikely to be associated with mid-trimester cervical shortening. Further research is needed to understand the mechanism of PTB in HIV-infected women receiving ART.

2. “Table 2 and 2a: The length <30 mm is not needed since >25 mm is considered normal.” 

Response 2: We agree, and removed this line in the table. 

3. “The information about ART can be in the text of the manuscript and not needed in Table.”

Response 3: This data is available in the text, and was removed from the table as suggested.

4. “Please amend your current ethics statement to address the following concerns: 

a) Did participants provide their written or verbal informed consent to participate in this study?

b) If consent was verbal, please explain i) why written consent was not obtained, ii) how you documented participant consent, and iii) whether the ethics committees/IRB approved this consent procedure.”

Response 4: All volunteers for this study signed an approved written consent in Setswana or English (per language preference) prior to participation in the study. We clarified this in the manuscript on page 9 under “Ethics Approvals”.

5. “We note that you have indicated that data from this study are available upon request. PLOS only allows data to be available upon request if there are legal or ethical restrictions on sharing data publicly. For information on unacceptable data access restrictions, please see http://journals.plos.org/plosone/s/data-availability#loc-unacceptable-data-access-restrictions.”

Response 5: The de-identified dataset for this study, including the outcomes data, was uploaded to the PLOS ONE site as a supporting file. Please see the attached file.

6. “Commentary regarding mechanism of progesterone use in HIV infected women with normal cervical length appears out of place in this manuscript as this is an ultrasound study, not a therapeutic study. If the subjects in the trial did get vaginal progesterone and the authors think this confounded the preterm birth rate in the cohort, that should be explicitly said. Otherwise I would remove this section.”

Response 6: To limit any confusion, we removed the paragraph in the discussion about vaginal progesterone. At this time, vaginal progesterone is not offered in Botswana as part of a prevention strategy to decrease risk of preterm birth, regardless of HIV status. 

7. “The acceptability data is interesting, and should be highlighted in the abstract if there is word count to support it.”

Response 7: Thank you for acknowledging the importance of the transvaginal cervical length acceptability data. We highlighted the acceptability data in the abstract as you suggested.

---

## [Editor Report · Decision Letter 1]

10 Feb 2020

Mid-trimester cervical length not associated with HIV status among pregnant women in Botswana

PONE-D-19-28835R1

Dear Dr. Liff,

We are pleased to inform you that your manuscript has been judged scientifically suitable for publication and will be formally accepted for publication once it complies with all outstanding technical requirements.

With kind regards,

David J. Garry, DO FACOG

Academic Editor

PLOS ONE

---

## [Editor Report · Acceptance letter]

27 Feb 2020

PONE-D-19-28835R1 

Mid-trimester cervical length not associated with HIV status among pregnant women in Botswana 

Dear Dr. Liff:

I am pleased to inform you that your manuscript has been deemed suitable for publication in PLOS ONE. Congratulations! Your manuscript is now with our production department. 

With kind regards,

on behalf of

Dr. David J. Garry 

Academic Editor

PLOS ONE